# Melatonin Supplementation Attenuates the Pro-Inflammatory Adipokines Expression in Visceral Fat from Obese Mice Induced by A High-Fat Diet

**DOI:** 10.3390/cells8091041

**Published:** 2019-09-06

**Authors:** Talita da Silva Mendes de Farias, Regislane Ino da Paixao, Maysa Mariana Cruz, Roberta Dourado Cavalcante da Cunha de Sa, Jussara de Jesus Simão, Vitor Jaco Antraco, Maria Isabel Cardoso Alonso-Vale

**Affiliations:** Department of Biological Sciences, Institute of Environmental Sciences, Chemical and Pharmaceutical, Federal University of Sao Paulo, Diadema 09913-130, Brazil (T.d.S.M.d.F.) (R.I.d.P.) (M.M.C.) (R.C.d.C.d.S.) (J.d.J.S.) (V.J.A.)

**Keywords:** adipose tissue, leptin, adiponectin, Il-6, Mcp-1, Tnf-α

## Abstract

Obesity is defined as a condition of abnormal or excessive fat accumulation in white adipose tissue that results from the exacerbated consumption of calories associated with low energy expenditure. Fat accumulation in both adipose tissue and other organs contributes to a systemic inflammation leading to the development of metabolic disorders such as type 2 diabetes, hypertension, and dyslipidemia. Melatonin is a potent antioxidant and improves inflammatory processes and energy metabolism. Using male mice fed a high-fat diet (HFD—59% fat from lard and soybean oil; 9:1) as an obesity model, we investigated the effects of melatonin supplementation on the prevention of obesity-associated complications through an analysis of plasma biochemical profile, body and fat depots mass, adipocytes size and inflammatory cytokines expression in epididymal (EPI) adipose depot. Melatonin prevented a gain of body weight and fat depot mass as well as adipocyte hypertrophy. Melatonin also reversed the increase of total cholesterol, triglycerides and LDL-cholesterol. In addition, this neurohormone was effective in completely decreasing the inflammatory cytokines leptin and resistin in plasma. In the EPI depot, melatonin reversed the increase of leptin, Il-6, *Mcp-1* and *Tnf-α* triggered by obesity. These data allow us to infer that melatonin presents an anti-obesity effect since it acts to prevent the progression of pro-inflammatory markers in the epididymal adipose tissue together with a reduction in adiposity.

## 1. Introduction 

Obesity results from an imbalance between energy consumption and energy expenditure, promoting an abnormal or excessive accumulation of fat in various regions of the body. Associated with the fat increase, a chronic low-grade inflammation that contributes to systemic metabolic disorders such as dislipidemia, hypertension, nonalcoholic fatty liver diseases, steatohepatitis, cardiovascular diseases, type-2 diabetes and even some cancers has been observed.

Despite the fact that the molecular mechanisms that associate obesity with higher incidences of these diseases are not yet fully defined, evidence indicates that white adipose tissue(s) is one of the first tissues to develop inflammatory responses in obesity conditions, as evidenced by the activation of classical proinflammatory pathways, exacerbated infiltration of macrophages, neutrophils and lymphocytes and a variety of pro-inflammatory mediators secretion [1,2,3]. It is now well established that white adipose tissue (WAT) is not only involved in energy storage but also functions as an endocrine organ that secretes various bioactive substances or adipo (cyto)kines such as leptin, adiponectin, tumor necrosis factor-alfa (TNF-α), interleukin-6 (IL-6), resistin and macrophage chemoattractant protein 1 (MCP-1), which are known to be involved in a wide range of physiological processes [4]. These adipokines play an important role in the pathophysiological link between increased adiposity and cardiometabolic alterations [4,5]. Indeed, it is also now established that an imbalance of pro- and anti-inflammatory adipokines secretion by WAT due to the expansion of fat mass in obesity exerts potential effects on obesity-linked metabolic disorders [6,7]. Since obesity-associated increases in these adipokines present a great contribution to the development of a dysfunctional WAT, characterized by the infiltration of pro-inflamatory immune cells in this tissue and unresolved inflammation, in addition to inappropriate extracellular matrix remodeling and impaired angiogenesis [8], all of which lead to the development of chronic low grade inflammation [9].

High levels of leptin resulting from leptin resistance and reduced adiponectin levels have been associated with obesity pathophysiology disorders [10]. On the other hand, insulin resistance has been associated with leptin resistance and a reduction of plasma adiponectin, which is reverted by the simultaneous administration of leptin and adiponectin [11,12]. Indeed, a dysregulated expression of these (and others) adipokines by WAT that occurs in obesity is one of the most important components that predisposes one to insulin resistance and that is predictive of the development of metabolic syndrome [13,14]. This occurs because a lot of adipokines interact with the insulin pathway and interfere in glucose and lipid metabolism. 

Evidence in the literature indicates that melatonin has a modulatory effect on energy metabolism [15,16,17], insulin secretion and insulin action, as well as the glucose end lipid metabolism of adipose tissue from rats [18,19,20,21]. Therefore, we hypothesized that the melatonin used as a therapeutic strategy could be used as a way to improve the low-grade inflammation observed in obesity conditions. According to some studies, the use of melatonin, a neurohormone produced by the pineal gland only in the night phase and which is responsible for the synchronization of innumerable physiological effects, is related to beneficial effects on the control of obesity and its complications [22,23]. Additionally, chronobiological melatonin aspects and their interrelationship with cytokines produced by adipocytes such as leptin and adiponectin have been evaluated [10,24] and promising results in the prevention and control of complications caused by obesity have been suggested.

Though some studies have evaluated the effects of melatonin and/or pinealectomy on the repercussions of adiponectin and leptin gene expression [19,24], there are a lack of studies related with the role of melatonin on the expression of these and other adipokines involved in the inflammatory response and insulin signaling in WAT by fat depots. Thus, we herein investigate whether melatonin supplementation could prevent the characteristic increase of pro-inflammatory adipokines produced by epididymal WAT during the development of obesity in mice.

## 2. Materials and Methods

### 2.1. Animals and Melatonin Supplementation

The study was performed according to protocols approved by the Ethics Committee of the Federal University of São Paulo (CEUA 5998280515). Eight-week-old male C57BL/6 mice obtained from the Center for Development of Experimental Models (CEDEME), Federal University of São Paulo, were housed at 3 mice per cage in a room with light–dark cycle (12-h light, 12-h dark cycle, lights on at 0600) and temperature of 24 ± 1 °C. Mice were divided into three groups: (a) Control (low fat) diet (CO), (b) high-fat diet (Obese), and (c) high-fat diet supplemented with melatonin 1 mg/kg (Obese + Mel). The CO diet contained 76% carbohydrate, 15% protein and 9% fat, and the high-fat diet (HFD) contained 26% carbohydrate, 15% protein, and 59% fat, in % kcal. Lard and Soybean oil (9:1) was used as fat source The detailed composition of the diet and energy distribution was provided in our previous study [25].

During obesity induction, the animals were supplemented with melatonin (1 mg/kg) [26] in drinking water during the dark phase, daily, for 10 weeks. Body weight and food intake were measured weekly. After 10 weeks of the experimental protocol, 12-hour fasted mice were killed by cervical dislocation, which occurred between 9am and 11am, after isoflurane anesthesia. Blood samples were centrifuged at 1500 rpm for 20 min at 4 °C, and serum were stored at −80 °C. Adipose depots were collected and weighted, and epididymal adipose fat (EPI) was processed as described below.

### 2.2. Glucose and Insulin Tolerance Tests

An oral glucose tolerance test (oGTT) and an insulin tolerance test (ITT) were evaluated after a 6-hour fast. For oGTT analysis, we administrated by gavage a 20% glucose 20% solution (1 g/kg b.w.). The blood glucose measurements were performed at 0, 15, 30, 45, 60 or 90 min. For ITT, animals were injejected intraperitoneally. with insulin (Humulin R, Lilly, 0.75 UI/kg b.w.), and glucose measurements were performed at 0, 10, 20, 30, 40, 50 or 60 min after injection. In both tests, blood samples were collected from the tail vein. This method was not stressful, as indicated by the low basal levels of the stress hormone corticosterone. oGTT and ITT were determined by using a glucometer (One Touch Ultra, Johnson and Johnson, New Brunswick, NJ, USA). The assays were always performed in all groups concomitantly in order to avoid any interference in the obtained results.

### 2.3. Adipocyte Isolation

Adipocyte isolation was performed as previously described [27] with slight modifications [28]. Briefly, Epi fat pads were diced in small fragments in a flask containing 4 mL of DMEM supplemented with HEPES (20 mM), glucose (5 mM), bovine serum albumin (BSA, 1%), and collagenase type II (1 mg/mL) at pH 7.4 and incubated for approximately 40 min at 37 °C in an orbital shaker. Isolated adipocytes were filtered through a plastic mesh (150 µm) and washed three times in a fresh buffer without collagenase. After washing and brief spinning, the medium was thoroughly aspirated, and adipocytes were harvested. Aliquots of isolated adipocytes suspensions were placed in a microscope slide, and 6 fields were photographed under an optical microscope (×100 magnification) coupled to a microscope camera (AxioCam ERc5s; Zeiss, Oberkochen, Germany), and mean adipocyte volume (4/3 × π × r^3^) was determined by measuring 100 cells using AxioVision LE64 software.

### 2.4. Blood Measurements

Triacylglycerol (TG) [29], fasting glucose, total cholesterol (TC), LDL-cholesterol [30], and HDL-cholesterol levels [31] were determined by colorimetric assays (Labtest Diagnostics, Lagoa Santa, MG, Brazil). 

### 2.5. RNA Extraction and Quantitative Real-Time Polymerase Chain Reaction (qPCR)

Total RNA was extracted from an EPI depot, reverse transcribed, and destined for quantitative PCR analysis as previously described [25]. An analysis of real-time PCR data was performed using the 2^−ΔΔC^_T_ method [32]. Data are expressed as the ratio between the expression of the target gene and housekeeping gene (18S gene). Primers used are presented: *Adipoq* (5’-3’sense: GCAGAGATGGCACTCCTGGA; 5’-3’antisense: CCCTTCAGCTCCTGTCATTCC), *Tnf-α* (5’-3’sense: CCCTCACACTCAGATCATCTTCT; 5’-3’antisense: GCTACGACGTGGGCTACAG), *Il-6* (5’-3’sense: TTCTCTGGGAAATCGTGGAAA; 5’-3’antisense: TCAGAATTGCCATTGCACAAC), *Lep* (5’-3’sense: CATCTGCTGGCCTTCTCCAA; 5’-3’antisense: ATCCAGGCTCTCTGGCTTCTG), *Mcp-1* (5’-3’sense: GCCCCACTCACCTGCTGCTACT; 5’-3’antisense: CCTGCTGCTGGTGATCCTCTTGT) and *18S* (5’-3’sense: GGCCGTTCTTAGTTGGTGGAGCG; 5’-3’antisense: CTGAACGCCACTTGTCCCTC). 

### 2.6. Adipokine Measurements

Lysates from the EPI depot and peripheral blood were used to perform the ELISA test. The concentrations of the adipokines IL-6, resistin, adiponectin and leptin were determined using specific commercially available DuoSet ELISA kits according to instructions supplied by the manufacturer (R&D Systems, Minneapolis, MN, USA; Catalog numbers DY406, DY1069, DY1119, DY 498, respectively). The concentrations of the cytokines were expressed in ng per 100 mg of tissue or in ng/ml, as indicated.

### 2.7. Statistical Analysis

Data are presented as mean ± SEM. A one-way ANOVA and a Tukey post-test were used for the comparison between groups. GraphPad Prism 5.0 software (GraphPad Software, Inc., San Diego, CA, USA) was used for analysis. The level of significance was set at *p* < 0.05. 

## 3. Results

### 3.1. Melatonin Supplementation Decreased Body Mass (BM), Adipose Depots Mass, Adipocytes Hypertrophy and Blood Biochemical Parameters Triggered by HFD-Induced Obesity

After 10 weeks of diet-induced obesity (DIO), it was found that the HFD was efficient in increasing the body mass of the animals (between approximately two- and five-fold). Concerning food, calories, and fat intake, as compared to CO diet, mice fed with the HFD presented a reduction (by 44%, *p* < 0.05) in food intake but an increase (by three-fold, *p* < 0.05) in fat intake, whereas a slight reduction was observed in calorie (by 20%) and water (by 22%) intake. However, mice that received the HFD associated with melatonin supplementation presented a lower body mass gain (between approximately three- and eight-fold compared to the control group). Though the mice supplemented with melatonin gained less body mass, when the food intake was mesured, both the Obese and Obese + Mel groups presented the same pattern of food, calories and fat consumption (Figure 1A and Table 1).

Corroborating the lower body mass gain, we observed that the adipocyte size from the visceral (epididymal—EPI) region of the Obese + Mel group was 42% smaller than the obese group, thus preventing the hypertrophy triggered by the HFD (Figure 1B,C). In the same way, a significant reduction was observed in the EPI and inguinal (ING) depots mass from the animals supplemented with melatonin. The retroperitoneal (RP) and the brown fat (BAT) depots’ mass did not present a significant reduction (Table 1).

The analysis of glucose and lipids serum concentrations indicated that the HFD significantly increased fasting glucose (21%), triglycerides (55%), total-cholesterol (52%) and LDL-cholesterol (60%) in the Obese group when compared to the Control group. This increase was prevented in animals that were supplemented with melatonin (Obese + Mel group), since the fasting glucose levels remained similar to the control group, and the serum levels of triglycerides, total cholesterol and LDL-cholesterol were reduced (a reduction of 26%, 21%, and 23%, respectively, in relation to the obese group). There were no significant differences in serum HDL-cholesterol between the groups (Table 1). 

### 3.2. Melatonin Supplementation Did Not Alter Glycemic Curve after Glucose and Insulin Tolerance Test (GTT and ITT Test)

After glucose load, both groups receiving the HFD (Obese and Obese + Mel) presented higher blood glucose levels (Figure 2A). The HFD groups showed lower responsiveness to insulin compared to the control group (Figure 2B). Melatonin supplementation did not alter both the oGTT and ITT tests—that is, it did not prevent the development of insulin intolerance or the response to oral glucose loading.

### 3.3. Melatonin Supplementation Decreased the Gene Expression of Inflammatory Cytokines on EPI Depot

Whereas obesity is accompanied by a low-grade systemic inflammation, we evaluated the gene expression of the main adipokines produced in the visceral adipose depot from mice under an obesity condition. It was found that the HFD significantly increased the gene expression of *Lep, Il-6, Mcp-1* and *Tnf-α* ((Figure 3A–D) compared to the Control group. Melatonin was able to prevent some of these effects, since the Obese + Mel group presented a significant reduction in the expression of *Lep* (50%) and *Mcp-*1 (55.8%) in relation to the Obese group. Moreover, the expressions of *Il-6* and *Tnf-α* in the EPI depot of mice supplemented with melatonin were partially prevented (44.6% and 44.8%, respectively), when compared to the Obese group. No change was observed in the *Adipoq* gene expression (Figure 3E). 

### 3.4. Melatonin Supplementation Reduced the Protein Expression of Inflammatory Cytokines on EPI Depot and Peripheral Bood

The protein expression analysis by ELISA corroborated the data presented by the gene expression analysis. *Lep* expression in the EPI depot was significantly increased in the Obese group compared to the control group (82%, *p* < 0.05). In contrast, the Obese + Mel group showed a reduction by 30% (*p* < 0.05) compared to the Obese group (Figure 4A). Thus, melatonin supplementation partially prevented this increase, indicating its important action in reducing these adipokine levels. In plasma, we observed an even more pronounced effect, where the Obese group showed an increase in plasma leptin of approximately three-fold compared to the Control group, and melatonin supplementation reduced these levels by 28% (Obese vs Obese + Mel group, *p* < 0.05, Figure 4E).

Melatonin supplementation also prevented the increase of IL-6 protein in the EPI depot triggered by the HFD since the Obese + Mel group showed a reduction of 51% (*p* < 0.05) when compared to the Obese group (Figure 4B). No statistical differences in adiponectin and resistin protein levels were observed in this adipose depot (Figure 4C,D). However, adiponectin expression showed a tendency (*p* = 0.0629) to increase in the Obese + Mel group (39% increase compared to the Obese group).

Finally, the HFD increased the resistin expression in plasma by 34%. Melatonin supplementation partially prevented this effect, since the Obese + Mel group presented an increase of only 11% compared to the Control group (Figure 4E).

## 4. Discussion

In this study, we evaluated the effects of melatonin on WAT inflammatory aspects in obese mice induced by an HFD. Initially, we verified that the experimental model adopted for DIO was efficient, since we observed that the animals fed an HFD showed a greater gain of body mass (with greater adiposity) and an increase in fasting glucose, triglycerides, total cholesterol and LDL-cholesterol in plasma, glucose intolerance and insulin resistance, corroborating the results obtained previously by our group [25]. Melatonin supplementation was effective in preventing most of the alterations triggered by the HFD, significantly hampering the gain of body mass and preventing the dyslipidemia progression. 

There are, in the literature, some works showing the effect of melatonin decreasing body weight, and this hormone has therefore been considered as a possible therapeutic agent against obesity [33,34]. Herein, using a daily melatonin dose of 1 mg/kg (close to the endogenous physiological level), it was observed that the Obese + Mel group had an attenuation of their body mass gain in relation to the non-supplemented group. Taken together with a reduction in both the EPI and ING depots, melatonin suplemmentation performed a protective effect on the development of obesity and adiposity in animals fed an HFD.

Similarly to the findings reported here, using HFD-induced obese Wistar rats and supplementing them with melatonin (25 ug/ml) for 11 weeks, Rios Lugo et al. [34], observed a decrease in the body mass gain of these animals. Favero et al. [33], using a leptin-deficient (ob/ob) mouse supplemented with melatonin in drinking water (100 mg/kg) for eight weeks, also observed a decrease of approximately 5% in total body mass and a decrease in the weight of visceral and subcutaneous fat depots (53% and 41%, respectively) in animals supplemented with melatonin. 

In contrast, melatonin supplementation in humans (3 mg daily for three months) did not promote changes in the body mass gain or loss of these individuals [35]. In the same way, Nduhirabandi et al. [36], using obese rats treated with melatonin (4 mg/kg), did not observe any decrease in the body mass despite having presented a cardioprotective effect. These controversial data may be due to the different doses and methods of melatonin administration.

It is believed that melatonin’s effect on reducing body mass observed in rodents is probably mediated by central and peripheral target tissues, which leads to the synchronization of circadian rhythms and improved glucose uptake acting directly in adipocytes [24,34]. Here, we showed that dietary-induced obese animals supplemented with melatonin presented a significant reduction in fasting glycemia, indicating again the beneficial action of melatonin on glucose homeostasis, Corroborating these data, other studies employing pinealectomized animals have presented a significant reduction in the expression of the glucose transporter (GLUT 4), as well as glucose intolerance and insulin resistance, which were reverted by melatonin treatment [16,37]. The direct action of melatonin on the adipocytes and myocytes metabolism has been reported. In isolated adipocytes from the inguinal fat of rats, melatonin inhibited isoproterenol-stimulated lipolysis through the inhibition of the cAMP-PKA pathway [38]. In C2C12 skeletal muscle cells, melatonin activates the IRS-1 insulin receptor, thus stimulating GLUT4 expression [39], and these data corroborate a reduced glucose uptake in the skeletal muscle from melatonin receptor-1 knockout mice [40].

It is known that dyslipidemia triggered by obesity plays an important role in the development and worsening of the inflammatory state [41]. Here, we have shown that melatonin supplementation was effective in partially preventing the increase in total cholesterol, LDL-cholesterol and triglycerides characteristic of DIO. Previous studies have already shown this action of melatonin on lipid homeostasis [42,43,44]. Wistar rats induced to diabetes by streptozotocin and treated with melatonin (10 mg/kg and 20 mg/kg) i.p. for two weeks also showed significant reductions in TG, TC and LDL-cholesterol levels [45]. However, in another study [46], C57Bl/6 mice induced to obesity by an HFD and treated with melatonin (10 mg/kg) for 12 weeks presented a reduction in body mass and LDL levels but not in TG levels. Considering melatonin supplementation in humans, it was observed that after three months of treatment, the daily use of melatonin (3 mg) was effective in significantly lowering TC and TG levels [35]. It is important to note that dyslipidemia is frequently observed in obese and/or diabetic individuals and that high plasma concentrations of total cholesterol and LDL-cholesterol are associated with an increased risk of cardiovascular disease [47,48]. On the other hand, the effects of melatonin on the cardiovascular system are well known. The removal of circulating melatonin causes hypertension in rats, and melatonin replacement prevents or reduces this effect [49,50]. Thus, the attenuation of serum triglyceride levels, total cholesterol and LDL-cholesterol reported here in animals treated with melatonin suggests a role for melatonin in the atherosclerosis prevention, one of the main complications of obesity.

There is a positive correlation between the increase in visceral WAT and hypertension, dyslipidemia, fasting glucose, non-alcoholic steatosis, age, and gender [51,52]. Thus, visceral obesity leads to increased risk of insulin resistance and cardiovascular disease. This fat depot displays a higher production of inflammatory cytokines in obese individuals [53,54]. 

We observed that the melatonin supplementation reduced the mass of the EPI depot by preventing cell hypertrophy because the adipocyte volume was reduced by 42%. Favero et al. [33], through histological and morphometric observations in WAT of animals supplemented with melatonin, observed that adipose depots from non-obese animals are composed of smaller and regular adipocytes, whereas in obese animals the adipocytes are larger and have a wide lipid droplet with presence of inflammatory infiltrate, macrophages and monocytes with degranulation signs which characterize the inflammatory tissue state (wherein there is a greater adipokines pro-inflammatory expression). Taken together, we suggest that melatonin contributes to the prevention of the inflammatory process in the visceral WAT (here represented by the EPI depot) because prevented the adipocytes hypertrophy.

As expected, we observed a significant increase in the expression of genes encoding proinflammatory cytokines, such as *IL-6, Lep, Mcp-1*, and *Tnf-α* in the EPI depot of animals with DIO. Melatonin fully reversed the increase of *Lep* and *Mcp-1* and partially reversed *Il-6* and *Tnf-α* gene expression. Melatonin supplementation also prevented an increase of leptin and IL-6 protein expression in the EPI depot triggered by the HFD. Moreover, melatonin was effective in decreasing the levels of leptin and resistin in plasma of animals induced to obesity by an HFD. 

In the subcutaneous adipose tissue of ob/ob mice, immunofluorescence analyses also revealed melatonin’s effects in reducing the expression of TNF-α, resistin, and visfatin, as well as in increasing the expression of adiponectin and its receptors [33]. In the liver of obese mice, Sun et al. [46] found that melatonin treatment also reduced the expression of proinflammatory markers *Tnf-α, Il-1β*, and *Il-6*. The same downregulation of these markers was observed in the liver of senescence accelerated prone male (SAMP8) mice [55]. In Wistar rats induced to obesity by an HFD, the plasma analyses demonstrated that melatonin attenuated the increase of the leptin observed in obese animals [34]. 

It is important to emphasize that the increase of these cytokines, such as leptin and IL-6, establishes a pathophysiological link between increased adiposity and obesity-associated cardiovascular disease, insulin resistance, type 2 diabetes, hypertension, and dyslipidemia [4,5,9]. On the other hand, obesity due to the ingestion of an HFD may be a consequence of a desynchronization in the biological rhythms of important metabolic processes [56,57] Consequently, the obese phenotype may be originated from this circadian desynchronization. Corroborating this, the clock genes and adipocytokines show circadian rhythmicity. The dysfunction of these genes is involved in the alteration of these adipokines during the development of obesity. Desynchronization between the central and peripheral clocks by an altered diet composition can lead to the uncoupling of peripheral clocks from the central pacemaker and to the development of metabolic disorders, leading to obesity. While CLOCK expression levels are increased with HDF-induced obesity, peroxisome proliferator-activated receptor (PPAR) alpha increases the transcriptional level of brain and muscle ARNT-like 1 (BMAL1) in obese subjects. Consequently, the disruption of clock genes results in dyslipidemia, insulin resistance and obesity [58]. Since melatonin is recognized by its important synchronization of diurnal and circadian rhythms [59]**,** the supplementation with melatonin, respecting its physiological pattern of secretion (exclusively at night as performed in this work), is an important synchronizer to break changes triggered by diet [59] and could prevent changes observed in the obese phenotype.

Other assays need to be performed so that we can conclude in more detail how melatonin acts to reduce proinflammatory cytokines in adipose tissue. However, based on the facts that chronic inflammation in visceral WAT is one of the first steps in triggering obesity-associated diseases and that supplementation with melatonin for 10 weeks revealed significant effects in reducing body mass gain, adiposity and visceral adipocytes hypertrophy, plasma lipids and fasting glucose, as well as the expression of proinflammatory markers in visceral adipose tissue, we can infer here that melatonin must be considered to be a reliable therapeutic agent for the treatment of obesity. 

## Figures and Tables

**Figure 1 cells-08-01041-f001:**
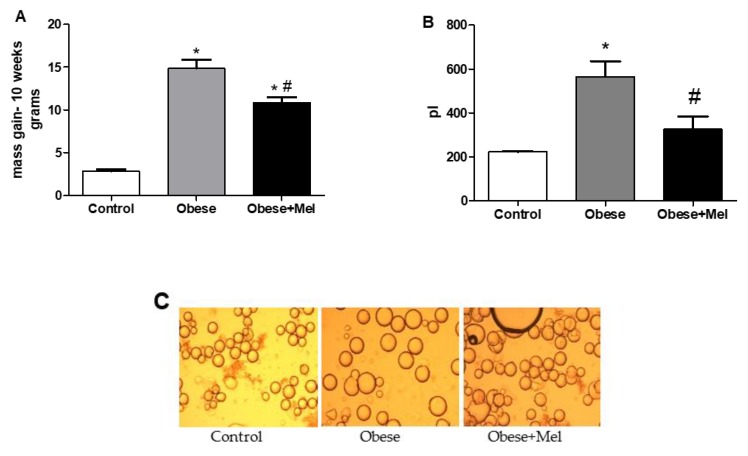
Effects of a high-fat diet (HFD) and melatonin supplementation (Mel, 1 mg/kg b.w., diluted in drinking water, daily, for 10 weeks) on body weight and adipocytes size. (**A**) Mice body mass gain at the end of experimental protocol. (**B**) Volume (in picoliters) of epididymal (EPI) isolated adipocytes. (**C**) Isolated EPI adipocytes photographed under optic microscope (×100 magnification). Adipocyte volume (4/3 × π × r^3^) was determined by measuring 100 cells per animal (six fields for each slide). Results were analyzed by a one-way ANOVA and a Tukey post-test. Values are mean ± SEM (Control *n* = 21; Obese *n* = 17; Obese + Mel *n* = 20). **p* < 0.05 vs. control; #*p* < 0.05 vs. obese.

**Figure 2 cells-08-01041-f002:**
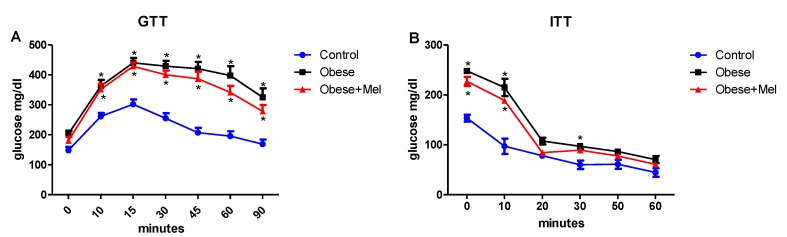
Effects of the high-fat diet (HFD) and melatonin supplementation (Mel, 1 mg/kg b.w., diluted in drinking water, daily, for 10 weeks) on glucose and insulin tolerance tests (GTT and ITT, respectively) in mice. (**A**) GTT or glucose concentration versus time after administration of glucose (2 g/kg b.w.); (**B**) ITT or glucose decay curve versus time after insulin administration (0.75 mU/g b.w.). Values are mean ± SEM (Control *n* = 11, Obese *n* = 7; Obese + Mel *n* = 11). **p* < 0.05 versus Control.

**Figure 3 cells-08-01041-f003:**
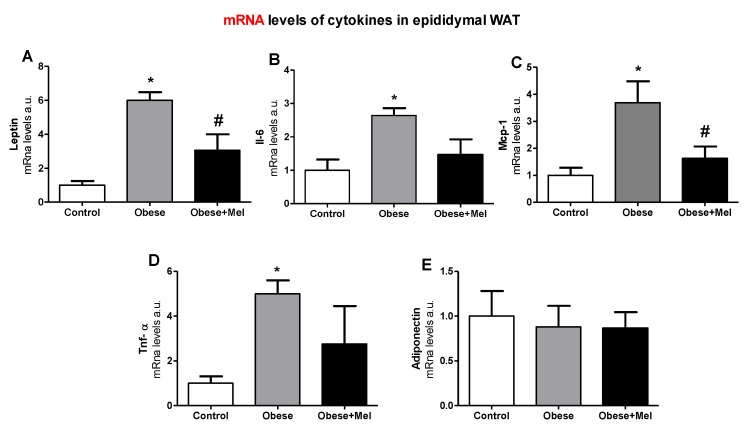
Effects of the high-fat diet (HFD) and melatonin supplementation (Mel, 1 mg/kg b.w., diluted in drinking water, daily, for 10 weeks) on mRNA levels of genes related to inflammation in epididymal (EPI) WAT from mice. (**A**) mRNA levels of *Lep*; (**B**) mRNA levels of *Il-6*; (**C**) mRNA levels of *Mcp-1*; (**D**) mRNA levels of *Tnf-α*; (**E**)mRNA levels of *Adipoq*. Results were analyzed by a one-way ANOVA and a Tukey post-test. Values are mean ± SEM (Control *n* = 9; Obese *n* = 8; Obese + Mel *n* = 10). **p* < 0.05 vs. Control; #*p* < 0.05 vs. Obese.

**Figure 4 cells-08-01041-f004:**
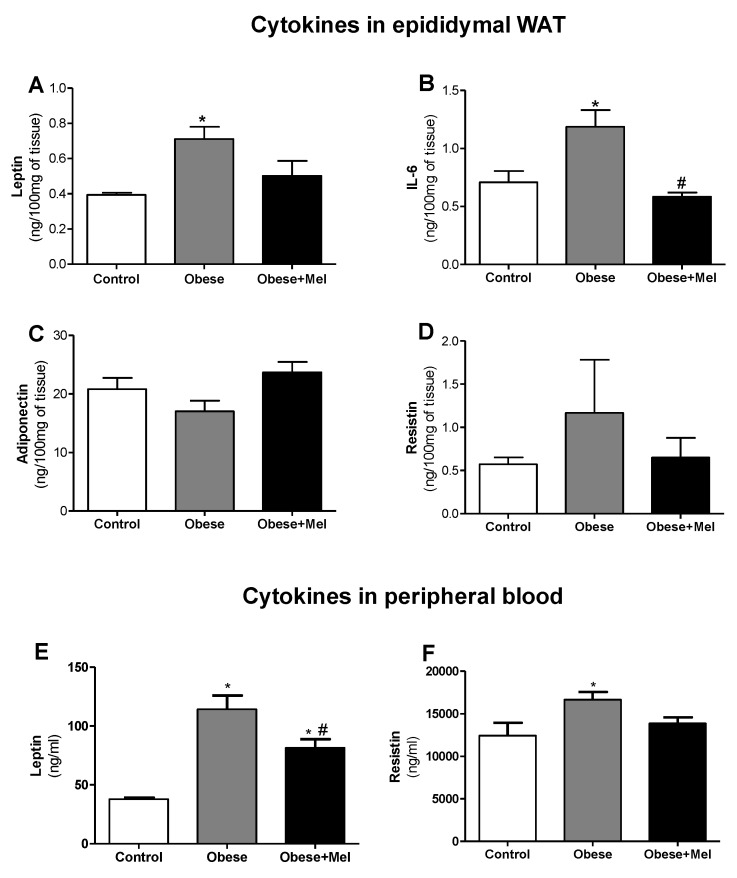
Effects of the high-fat diet (HFD) and melatonin supplementation (Mel, 1 mg/kg b.w., diluted in drinking water, daily, for 10 weeks) on ELISA analysis of cytokine levels in lysates from epididymal (EPI) WAT and peripheral blood from mice. (**A**) Leptin expression on the EPI depot, (**B**) Il-6 expression on the EPI depot, (**C**) adiponectin expression on the EPI depot, (**D**) resistin expression on the EPI depot, (**E**) leptin expression on peripheral blood, and (**F**) resistin expression on peripheral blood. Results were analyzed by a one-way ANOVA and a Tukey post-test. Values are mean ± SEM (Control *n* = 10; Obese *n* = 9; Obese + Mel n = 11). **p* < 0.05 vs. Control; #*p* < 0.05 vs. Obese.

**Table 1 cells-08-01041-t001:** Body mass (BM), food intake, organ weights and blood biochemical parameters after 10 weeks of a high-fat diet feeding and melatonin suplementation in mice.

	Control	Obese	Obese + Mel
Initial BM (g)	22.70 ± 0.25	23.47 ± 0.29	22.56 ± 0.31
Final BM (g)	25.56 ± 0.30	38.22 ± 1.15*	33.44 ± 0.66*#
Water intake (ml/day/mice)	3.97 ± 0.20	3.08 ± 0.11*	3.20 ± 0.08*
Food Intake (g/day/mice)	4.76 ± 0.30	2.45 ± 0.20*	2.34 ± 0.22*
Calories intake	18.09 ± 0.32	13.08 ± 0.30*	12.51 ± 0.34*
Fat intake (g/day/mice)	0.43 ± 0.03	1.44 ± 0.12*	1.38 ± 0.13*
Relative ING weight (g/100g BM)	1.52 ± 0.07	4.67 ± 0.21*	3.86 ± 0.16*#
Relative EPI weight (g/100g BM)	2.04 ± 0.13	6.51 ± 0.20*	5.54 ± 0.17*#
Relative RP weight (g/100g BM)	0.45 ± 0.04	1.84 ± 0.05*	1.70 ± 0.04*
Relative BAT weight (g/100g BM)	0.25 ± 0.01	0.34 ± 0.02*	0.31 ± 0.01*
Fasting blood glucose (mg/dl)	175.64 ± 12.72	243.53 ± 8.20*	202.08 ± 13.61#
Total Cholesterol (mg/dl)	173.57 ± 13.19	265.50 ± 18.65*	209.30 ± 18.38
Triglycerides (mg/dl)	83.04 ±5.34	129.12 ±14.55*	94.87 ± 9.08
HDL (mg/dl)	69.56 ± 6.40	82.28 ±15.24	84.13 ± 11.62
LDL (mg/dl)	92.29 ± 9.74	148.34 ± 16.90*	114.22 ± 17.92

Results were analyzed by a one-way ANOVA and a Tukey post-test. Values are mean ± SEM. Control *n* = 20; Obese *n* = 17; Obese + Mel *n* = 21 to body mass, food, and fat intake and depots weight; Control *n* = 10; Obese *n* = 9; Obese + Mel *n* = 9 to blood biochemical parameters). **p* < 0.05 vs. Control; #*p* < 0.05 vs. Obese.

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
