# Peer review of "Melatonin Supplementation Attenuates the Pro-Inflammatory Adipokines Expression in Visceral Fat from Obese Mice Induced by A High-Fat Diet"

_cells, 2019, doi:10.3390/cells8091041_

Round 1
Reviewer 1 Report
The topic is interesting but this version of the manuscript is lacking of important experiments necessary to validate the scientific hypothesis. Main criticisms are the following: 1.Abstract: Please insert the type of high fat diet or the energy provided by fat; insert also the sex of mice and the type of visceral depot analyzed (if inguinal, retroperitoneal or epididymal). 2.Introduction-Enlarge details on cytokines types and pathways involved.Authors cited 1-3 as References but they diid not explain their content. Line 65 please insert "epididymal" WAT instead of "visceral" WAT.3.Materials and methods-Please insert the number of mice for each experimental group. More experiments are required such as:ELISA analysis of cytokines; Immunostaining of selected cytokines in epidydymal WAT; Analysis of pyroptosis and mitochondria in adipose tisse depots. 4.results-Blood datra are incomplete and ITT and HOMA-IR may be added.Why Authors analyzed only epididymal WAT but reported also weight dat in Table 1 of other fat depots like visceral WAT and BAT? 5.Discussion-Insert at line 164 "epididymal" . Authors discussed interesting previous papers but data in this study were restricted to blood analysis and mRNA expression of cytokines in epididymal fat. The efficacy of melatonin supplementation in cardiovascular or metabolic diseases may be only inferred. Moreover, the text from line 260 to line 268 is too speculative because here Authors did not analyze mitochondria nor pyroptosis. Finally, conclusions are interesting but not supported by sufficient experimental data.
Author Response
We thank the Reviewer for all criticism and suggestion that are very appropriate. We incorporated all requested changes in the new version of the manuscript (highlighted in red text). Concerning the new experiments requested, we thanks the Reviewer for emphasizing the importance of investigating by ELISA the cytokines in EPI WAT to understand whether the changes in mRNA levels induced by melatonin are translated in changes in protein content. As depicted in Figure 4A and 4B, we found that the treatment of obese animals with Mel significantly decreased the Leptin and IL-6 protein levels in visceral (EPI) adipose depot. Furthermore, we have also performed the analysis of the cytokine by ELISA in plasma and we have also added the results (see Figure 4E and 4F). We are confident that the results obtained now further strengthen the conclusion of the work and the quality of the manuscript have been substantially improved.
About the ITT requested, we had already performed the ITT and GTT tests in the animals, but as we had not seen a statistically relevant response due to Mel treatment, we had chosen not to present. The data has been now incorporated into the new version of the article (Figure 2)
Concerning the question: “Why authors analyzed only epididymal WAT but reported also weight data in table 1 of other fat depots like visceral WAT and BAT?”
Only epididymal adipose tissue has been analyzed in this work and others (see references below), since it is recognized as a visceral tissue that responds effectively to changes caused by the high-fat diet in mice and rats in terms of inflammation, release of inflammatory mediators, insulin resistance, etc, more than any other adipose depots, such as ING subcutaneous. Alterations in the visceral WAT promoted by obesity are characterized by infiltration of pro-inflammatory immune cells in this tissue and unresolved inflammation, besides to inappropriate extracellular matrix remodeling and impaired angiogenesis [Crewe C, An YA, Scherer PE. J Clin Invest. 2017;127:74–82.doi:10.1172/JCI88883], which leads to the development of chronic low-grade inflammation [McNelis & Olefsky, Immunity. 17;41(1):36-48. doi: 10.1016/j.immuni.2014.05.010].
The weights of the other depots were incorporated into the table 1 in order to reinforce the efficacy of melatonin supplementation in preventing body mass gain and adiposity in animals consuming a high-fat diet since these depots also presented a reduced mass.
Some references:
Potential Anti-obesogenic Effects of Ginkgo biloba Observed in Epididymal White Adipose Tissue of Obese Rats. Front Endocrinol. 2019 May 10;10:284. doi: 10.3389/fendo.2019.00284. eCollection 2019.
Clcn3 deficiency ameliorates high-fat diet-induced obesity and adipose tissue macrophage inflammation in mice. Acta Pharmacol Sin. 2019 Jun 5. doi: 10.1038/s41401-019-0229-5
Cannabinoid Receptor 1 Blockade Attenuates Obesity and Adipose Tissue Type 1 Inflammation Through miR-30e-5p Regulation of Delta-Like-4 in Macrophages and Consequently Downregulation of Th1 Cells. Front Immunol. 2019 May 10;10:1049. doi: 10.3389/fimmu.2019.01049. eCollection 2019.
Reviewer 2 Report
The paper of Mendes de Farias et al shows a good study on the influence of melatonin in drinking water onto a HFD model in mice and some parameters of obesity as well as on some agents of inflammation (adipokines).
The paper experimental is well done and clear. The conclusion should be slightly more focused on the results provided by the experiments, rather than a too long discussion on the various roles of the various gene products. A few line on the meaning of the possible transposition of the exp to Human would be wise and welcome.
Minor: line 11 and line 148 : Is obesity a disease ? Wouldn’t it be a state leading to disease-like complications (such as diabetes)???
Minor: line 14: melatonin is NOT related to the improvement etc.. It causes it, no?
Minor: Line 19: Melatonin prevented the body and fat depots weight …: this should be rewritten. For example: “Melatonin prevents gain in weight and fat deposit”???
Minor: line 23-24: melatonin is a potential ant-obesity efficacy???? The sentence is not odd, and I am not sure it is what the authors wanted to say??
Minor: line 73: please, indicate roughly the amount per animal (~20 µg?) and translate that to human (80 mg???). Furthermore, it is hard to know what exactly is the amount of melatonin delivered per animal, as 1/ they are alone in cage (??) and 2/ it is not reported the amount of liquid proposed per cage (???) The water consumption should be indicated in Table 1.
Minor Line 115: please define BM at first use, not at the second one (Line 142)
Minor: line 123: the adipocyte size is lower. This is not correct: smaller??
Minor: Figure 1C: how many fields in the microscope were analyzed? Please, give details in the legend of the figure.
Minor Line 150: HDF? HFD!!
Minor: lines 151-156: no data on the amount of proteins (rather than the gene) for the pro-inflammatory compounds?
Major: the discussion is too long. I strongly recommend cutting a third, at least, of it. The case is straight forward, so discussion is misleading the reader. Be more concise on the conclusions brought by your report. It is maybe not so important to recall the reader the role of all those proteins into the inflammation process.
Author Response
Thank you very much indeed for the appropriate criticisms and corrections. The text was changed to incorporate your recommendations (highlighted in red text).
Minor: line 11 and 148: “Is obesity a disease? Wouldn’t it be a state leading to disease-like complications??“
Thanks for the valuable comment. We have now amended the text and we believe it can be successfully addressed. In fact, the study of obesity is very complex. However, in 2013 delegates from the American Medical Association (AMA) voted to recognize obesity as a disease. This decision was supported by other important associations such as the American Association of Clinical Endocrinologists, Endocrine Society, American College of Cardiology and American Heart Association (PMID: 30323513).
Clinically, obesity is defined as a condition of abnormal or excessive fat accumulation in adipose tissue, of sufficient extent to produce adverse health consequences [Hebebrand J, et al. A proposal of the European association for the study of obesity to improve the ICD-11 diagnostic criteria for obesity based on the three dimensions etiology, degree of adiposity and health risk. Obes Facts. 2017;10:284–307. doi:10.1159/000479208], including cardiometabolic alterations such as type 2 diabetes, hypertension, and dyslipidemia [Gomez-Ambrosi J, Catal_an V, Rodrıguez A, et al. Diabetes Care. 2014;37:2813–20. doi:10.2337/ dc14-0937; Freuhbeck G, Catal_an V, Rodr_ıguez A, et al. Sci Rep. 2017;7:2752. doi:10.1038/s41598-017-02848-0].
Minor: line 14: “melatonin is NOT related... It causes it, no?”
We agree with the reviewer. The correction was made.
Minor: line 19: Melatonin prevented the body and fat depots weight… this should be rewritten. For example: “Melatonin prevents gain in weight and fat deposit”???
The sentence has been rewritten.
Minor: lines 23-24: melatonin is a potential anti-obesity efficacy???? Thee sentence is not odd, and I am not sure it is what the authors wanted to say???
The sentence has been rewritten.
Minor: line73: “Indicate roughly the amount per animal (~20ug?) and translate that to human (80mg??Furthermore, it is hard to know what exactly is the amount of melatonin delivered per animal, as 1/they are alone in the cage(??) and 2/ it is not reported the amount of liquid proposed per cage (???) The water consumption should be indicated in Table 1”.
At the end of the experimental protocol, the animals weighed between 25 and 35 grams, so we can consider that they ingested between 0.025 and 0.035 mg of melatonin per day. To ensure this intake, the volume of water ingested was measured daily to adjust the offered dose. Extrapolating these values to humans, we can consider that an adult of 80 kg should ingest an amount of 80 mg of melatonin per day to have the same benefits.
Three animals were kept per cage. Although the reviewer is completely right regarding the accuracy of the amount of melatonin ingested in the drinking water, we chose not to use individual cages. The isolation of the animal entails other changes to the protocol since mice are sociable animals [Oksana Kaidanovich-Beilin O, Lipina T,1, Vukobradovic I, et al. Jove 2011, doi:10.3791/2473]. The average water intake was included in TABLE 1. We thank the reviewer for the very relevant point raised.
Minor: line 115: Please define BM at the first use, not at the second one (line 142)
The definition was made.
Minor: line 123: The adipocyte size is lower. This is not correct: smaller??
The correction was made.
Minor: Figure 1c: How many fields in the microscope were analyzed? Please, give details in the legend of the figure.
Aliquots of isolated adipocytes suspensions were placed in a microscope slide and 6 fields (for each animal) were photographed under an optical microscope (X100 magnification) coupled to a microscope camera (AxioCam ERc5s; Zeiss, Oberkochen, Alemanha), and mean adipocyte volume (4/3 x π x r3) was determined by measuring 100 cells using AxioVision LE64 software.
The details were included in the legend of the Figure.
Minor: line 150: HDF? HFD!!
The correction was made.
Minor: lines 151-158: no data on the amount of proteins (rather than the gene) for the pro-inflammatory compounds?
The data were added to the manuscript.
Major: “the discussion is too long...” Thanks for the relevant comment, which we believe can be successfully addressed. We have now reduced the discussion and take care to be more concise in the conclusion.
Reviewer 3 Report
In this manuscript da Silva Mendes de Farias and coworkers analyze the therapeutic potential of melatonin supplementation in diet-induced obesity in mice. They show that increased melatonin supplementation leads to decreased body weight gain and adiposity in mice on a high-fat diet (HFD). This is accompanied by decreased mRNA levels of leptin and pro-inflammatory cytokines in visceral adipose tissue. The data point at a potential role of melatonin as anti-obesity drug.
This is a very interesting short paper of high potential for clinical medicine. It also reads well. My main concern is the use of melatonin supplementation for all three treatment groups (NC and two HFD groups). As such the major difference between the obese and the obese+mel groups appears to be an increased dosing of melatonin. An HFD-no mel (neither in food nor in drinking water) and an NC-no mel group should be included to show how the absence or presence of melatonin affects DIO.
Further points:
When comparing the two obese groups it would be informative to know the real melatonin levels in the blood. How does the additional supplementation in the food alter the diurnal profile of melatonin in the blood? Could circadian disruption play a role in the observed metabolic phenotype? Table 1: food and fat intake should be presented as calories/day/mouse At which time of day were animals sacrificed and tissues sampled? Could some of the changes be explainable by alterations in diurnal rhythms? It is known that DIO affects circadian organization (PMID:17983587) and it is not unlikely that melatonin may interact with this effect. Why do cohort sizes vary so much between readouts (17-21 in Fig. 1; 9-13 in Fig. 2)? 2: for leptin and adiponectin it would be more informative to report blood levels rather than mRNA.
Author Response
Thank for the positive comments about our paper. About the relevant points raised, some experiments and information have been added to the text.
Comments:
“An HFD-no mel (neither in food nor in drinking water) group should be included to show how the absence or presence of melatonin affects DIO”
This protocol was performed since the experimental design consists of 3 groups:
1- Control group: animals were fed with a control diet (76% carbohydrate, 15% protein and 9% fat).
2- Obese group: animals were fed with a high-fat diet -HFD (26% carbohydrate, 15% protein and 59% fat)
3- Obese+Mel group: animals were fed with a HFD and received melatonin in the drinking water (only at night), at a dose of 1mg / kg.
Further points:
“When comparing the two obese groups it would be informative to know the real melatonin levels in the blood. How does the additional supplementation in the food alter the diurnal profile of melatonin in the blood?”
To demonstrate the actual levels of melatonin in mice plasma, it would be necessary to collect the blood in a dimly-lit environment, as melatonin is a highly photosensitive and easily degraded hormone. Unfortunately, we do not have these lab conditions to perform such a procedure, so we did not present this data, although we recognize its importance. In the case of C57bl mice, endogenous production is below the detection threshold by Elisa assays (DOI: 10.1080 / 07420528.2019.1624373), so we believe that for the adopted animal model, the supplementation of 1mg / kg is considered a pharmacological dose.
“Could circadian disruption play a role in the observed metabolic phenotype?”
Obesity due to the ingestion of a high-fat diet may be a consequence of desynchronization in the biological rhythms of important metabolic processes (doi: 10.1007/978-3-319-48382-5. Review; doi:10.1016/j.brainres.2008.12.071; doi.org/10.1016/j.cmet.2007.09.006). Consequently, the obese phenotype may be originated from this circadian desynchronization. Since melatonin is recognized by its important synchronizing on diurnal and circadian rhythms (Synchronizing effects of melatonin on diurnal and circadian rhythms. doi: 10.1016/j.ygcen.2017.05.013. Epub 2017 May 19. Review), the supplementation with melatonin (exclusively at night) could prevent the changes observed in the obese phenotype, that is, the desynchronization generated by the HFD intake.
“Table 1: food and fat intake should be presented as calories/day/mouse. At which time of the day were animals sacrificed and tissues sampled?
The data were included in Table 1 (highlighted in red text).
The animals were sacrificed between 9 am and 11 am, during which time the samples were collected. This information has been added to the text (highlighted in red text).
“Could some of the changes be explained by alterations in diurnal rhythms? It is known that DIO affects the circadian organization and it is not unlikely that melatonin may interact with this effect.”
Evidence in the literature indicates that the diet "per se" can alter the rhythmicity of some physiological processes (as cited above). The clock genes and adipocytokines show circadian rhythmicity. Dysfunction of these genes is involved in the alteration of these adipokines during the development of obesity. Desynchronization between the central and peripheral clocks by altered timing of food intake and diet composition can lead to uncoupling of peripheral clocks from the central pacemaker and the development of metabolic disorders. Metabolic dysfunction is associated with circadian disturbances at both central and peripheral levels and, eventual disruption of circadian clock functioning can lead to obesity. While CLOCK expression levels are increased with high fat diet-induced obesity, peroxisome proliferator-activated receptor (PPAR) alpha increases the transcriptional level of brain and muscle ARNT-like 1 (BMAL1) in obese subjects. Consequently, disruption of clock genes results in dyslipidemia, insulin resistance and obesity (Circadian Rhythms in Diet-Induced Obesity. doi: 10.1007/978-3-319-48382-5_2. Review.)
Moreover, despite the known changes in the rhythmic patterns of the processes studied in this work, the present work aimed to verify if the supplementation with melatonin would be able to improve or even block the alterations caused by the HFD ingestion. We do believe that the melatonin supplementation, respecting its physiological pattern of secretion, that is, offered to the animals only at night, is an important synchronizer to break the changes triggered by diet. (melatonin is recognized by its important synchronizing on diurnal and circadian rhythms; doi: 10.1016/j.ygcen.2017.05.013. Epub 2017 May 19. Review)
Another point we would like to emphasize is that the animals were always sacrificed at the same time, with a maximum interval of 2 hours (animals were euthanized intercalated between groups), avoiding changes in the circadian rhythm as a consequence of the time that mice were sacrificed.
“Why do cohort sizes vary so much between readouts? “
The analysis of body mass gain, water and food intake, weight of adipose depots and adipocytes size were performed for all lots of animals used in this paper. After animal sacrifice, several types of analyses were performed, where half of the samples were processed for RNA extraction and gene expression studies and the other half for protein extraction and ELISA. For blood biochemical measures, ITT and GTT analysis, only the second lot of animals were performed.
“For leptin and adiponectin, It would be more informative to report blood levels rather than mRNA.”
As suggested by the reviewer we have evaluated melatonin effects on some adipokines blood levels for this experimental protocol. As depicted in Figure 4E and 4F, we found that the treatment of obese animals with Mel significantly decreased the Leptin and Resistin plasma levels. Furthermore, we have performed the cytokines protein expression analysis by ELISA in visceral (EPI) adipose depot and we have also added the results (see Figure 4A, 4B, 4C and 4E). We are confident that the results obtained now further strengthen the conclusion of the work and the quality of the manuscript have been substantially improved. Unfortunately, we did not have enough time to wait for Adiponectin ELISA Kit import purchase (the kit we had has been finished).
The text was changed to incorporate your recommendations (highlighted in red text).
Round 2
Reviewer 1 Report
Authors ameliorated the manuscript but minor changes have still to be addressed before yhe full acceptance.
1.Line 95 Authors must indicate how glucose was administered if orally or i.p.
Lines 96-97 GTT method must be rewritten. At line 101 "glucose" word must be deleted.
2.Line 134 "WAT epididymal tissue" might be replaced by "EPI depot" and kept this term throughout the text.
3.Lines 151-152 How is it possible to introduce reduced calories by taking a high fat diet? How Authors estimated calories indicated in Table 1?
4. Check better for typing errors. At line 218 "moreover, the expression.." must be rewritten as "Moreover, the expressions..."
5. Figure 4A insert significance in obese + melatonin group.At lines 252-253 is the trend statistically significant? If so, please insert the symbol in Figure 4E.
6. Lines 295-297-How Authors discuss that melatoni was ineffective in GTT and ITT analysis but able to reduce fasting glucose reported in Table 1?
7.In the last sentence, at line 373 please insert the word "reliable" as follows "..that melatonin must be considered as a reliable therapeutic agent for the treatment of obesity."
Author Response
“Authors ameliorated the manuscript but minor changes have still to be addressed before the full acceptance.”
Response: We thank the Reviewer for all criticism and suggestion that are very appropriate. We incorporated all requested changes in the new version of the manuscript (highlighted in red text).
1. Line 95 Authors must indicate how glucose was administered if orally or i.p.
The indication was made.
Lines 96-97 GTT method must be rewritten. At line 101 "glucose" word must be deleted.
The sentence has been rewritten and the correction was made.
2. Line 134 "WAT epididymal tissue" might be replaced by "EPI depot" and kept this term throughout the text.
Done.
3.Lines 151-152 How is it possible to introduce reduced calories by taking a high-fat diet? How Authors estimated calories indicated in Table 1?
Response:
The detailed composition of the diet and energy distribution is provided in Table below:
Composition of the diets
|
Ingredients |
CO |
HF |
|
g/ Kg |
||
|
Starch |
465.7 |
115.5 |
|
Casein |
140 |
200 |
|
Dextrinized starch |
155 |
132 |
|
Sucrose |
100 |
100 |
|
Soybean oil |
4 |
35 |
|
Lard |
36 |
315 |
|
Cellulose |
50 |
50 |
|
Mineral mix |
35 |
35 |
|
Vitamin mix |
10 |
10 |
|
Cistine |
1.8 |
3 |
|
Choline bitartate |
2.5 |
2.5 |
|
kcal (%) |
||
|
Protein |
15 |
15 |
|
Carbohydrate |
76 |
26 |
|
Fat |
9 |
59 |
Based on food intake data and energy distribution from both (CO and HF) diets, the calories intake was calculated in kcal/day/animal. We sincerely apologize for not including this information in the table before. We have now amended adding the full information in table 1 in the revised text.
Concerning the comment “How is it possible to introduce reduced calories by taking a high-fat diet?”,
These same results were reported in other studies (de Sá et al., J Physiol. 2016 Nov 1;594(21):6301-6317. doi: 10.1113/JP272541; Masi et al., Sci Rep. 2017 Jun 21;7(1):3937. doi: 10.1038/s41598-017-04308-1; Westerterp-Plantenga et al., Annu Rev Nutr. 2009;29:21-41. doi: 10.1146/annurev-nutr-080508-141056; Martinez et al., Nat Rev Endocrinol. 2014 Dec;10(12):749-60. doi: 10.1038/nrendo.2014.175).
Protein, carbohydrates, and lipids are the components of foods that contribute to the energy supply (calories) to the diet. Dietary fat content is an important factor in energy balance, as lipid has twice as much energy per unit weight as protein or carbohydrate (Amuna & Zotor, Proc Nutr Soc. 2008 Feb;67(1):82-90. doi: 10.1017/S0029665108006058; McCrory et al., Physiol Behav. 2012 Nov 5;107(4):576-83. doi: 10.1016/j.physbeh.2012.06.012)
For a long time, caloric restriction was related to lower body mass gain, but the composition and quality of macronutrients present in the diet should be taken into consideration. The contribution of macronutrients to the promotion of weight loss in obese individuals has received increasing attention to determine whether calories derived from different sources have beneficial values for the body. Each macronutrient (its excess and its subtypes), more than the amount of calories ingested, affect metabolism, appetite, thermogenesis and weight loss, differently. (Martinez at al., Nat Rev Endocrinol. 2014 Dec;10(12):749-60. doi: 10.1038/nrendo.2014.175; Martins at al., J Nutr Biochem. 2018 May;55:76-88. doi: 10.1016/j.jnutbio.2017; Wilson et al., Mol Nutr Food Res. 2017 Jun;61(6). doi: 10.1002/mnfr.201600943).
We showed that, as compared to CO diet, mice fed with HFD presented a reduction (by 44%) in food intake but an increase (by 3-fold) in fat intake whereas a slight reduction was observed in calorie (by 20%). However, feeding efficiency (ratio of body weight gain and food consumption) and energy efficiency (ratio of body mass gain and energy consumption) were significantly increased (by 11- and 8-fold, respectively) in the HF group. Thus, we have seen that even eating less and totaling fewer calories, lipid intake potentially influences not only the increase in body mass but also all the negative metabolic consequences associated with obesity such as glucose intolerance and insulin resistance and dyslipidemia (leading to the onset of CVDs).
Check better for typing errors. At line 218 "moreover, the expression.." must be rewritten as "Moreover, the expressions..."
The correction was made.
5. Figure 4A insert significance in obese + melatonin group. At lines, 252-253 is the trend statistically significant? If so, please insert the symbol in Figure 4E.In Figure 4A, only the Obese group is significantly different from the Control (p≤ 0.05). The Obese+Mel group is not statistically different from either the Obese or the Control group. Considering this data, we conclude that melatonin supplementation partially prevented the increase of Leptin.
Concerning the lines 252-253, as we consider p<0.05 as statistically significant result, adiponectin expression showed only a tendency (p = 0.0629) to increase in the Obese+Mel group (39% of increase compared to Obese group).
Lines 295-297 -How Authors discuss that melatonin was ineffective in GTT and ITT analysis but able to reduce fasting glucose reported in Table 1?Type 2 DM usually results from varying degrees of insulin resistance and relative deficiency of insulin secretion, passing through intermediate stages that are called “altered fasting glucose” and “impaired glucose tolerance”. Any of the stages, preclinical or clinical, can lead in both directions, progressing to the diabetic state or reverting to normal glucose tolerance. What should be happening is a slight increase in insulin plasma and / or a slight improvement in its sensitivity in melatonin-treated animals, since previous work has shown that melatonin improves insulin action and secretion (Heo et al., J Pineal Res. 2018 Sep;65(2):e12493. doi: 10.1111/jpi.12493; Sun et al., Int J Endocrinol. 2018; 2018: 2304746. doi: 10.1155/2018/2304746; Yavuz et al., Acta Histochem. 2003;105(3):261-6; Picinato et al., J Pineal Res. 2002;33(3):172-177; Lima et al., Am J Physiol. 1998;275(6 Pt 1):E934-941.
Taken together, these effects, although subtle to statistically alter the GTT or ITT tests, reduced fasting hyperglycemia.
In the last sentence, at line 373 please insert the word "reliable" as follows "..that melatonin must be considered as a reliable therapeutic agent for the treatment of obesity."The correction was made.
Reviewer 3 Report
I thank the authors for considering my comments. The manuscript has improved. I still have concerns about the interpretation of the 1-timepoint data considering that we are looking at highly rhythmic parameters in most cases. I also still cannot understand why cohort sizes differ between different measurements and groups. While this is clear for behavioral (high n's) and molecular data (low n's) why do numbers change between groups and molecular readouts? One might suspect that certain data points were excluded from specific data sets. If that would be the case it would be very informative to know why this was done. Were outliers excluded? Were samples excluded for technical reasons?
Author Response
Response: We thank the Reviewer for the positive assessment of our paper.
In fact, adipokines present a rhythmic pattern of expression over 24 hours, We do believe if tissue collections were performed at another zeitgeber times (ZT) we would certainly find results different from those found here. It is also clear that a temporal study with tissue collections performed at different ZTs would point us to a more accurate data regarding circadian rhythm alterations and could thus infer whether melatonin supplementation adjusted for the dislocation promoted by the HF diet. However, this study aimed to verify whether or not melatonin supplementation could improve the changes induced by HF diet-obesity. The choice of the time for euthanasia was based on the animals' fasting time (12 hours). The night before, we allowed the animals to feed for 3 hours (from 6 pm to 9 pm) and at 9 pm the feed was removed and we kept the animals fasting for 12 hours.
Anyway, the question raised by the referee makes us think of a future project in which collections can be performed in different Zts to conduct a study of circadian analysis.
The “n” differs between the groups for 2 main reasons: 1. We lost some samples during the RNA extraction process and other samples did not amplify (PCR); 2. Some outliers have been excluded. We use a tool available at the following site: https://www.graphpad.com/quickcalcs/Grubbs1.cfm.
Only significant outliers were excluded. We use this tool since studies with animals show great biological variability